# End-to-End On-Device Quantization-Aware Training for LLMs at Inference Cost

## Abstract

Quantization is an effective technique to reduce the deployment cost of large language models (LLMs), and post-training quantization (PTQ) has been widely studied due to its efficiency. However, existing PTQ methods are limited by their inability to fine-tune model parameters and often suffer significant accuracy loss in low-bit scenarios. Quantization-aware training (QAT) provides a more principled solution, but its reliance on backpropagation incurs prohibitive memory costs, limiting its practicality for LLM deployment. To address these challenges, we propose ZeroQAT, a zeroth-order optimization-based QAT framework that supports both weight and activation quantization. ZeroQAT leverages forward-only gradient estimation to eliminate backpropagation, substantially reducing computational and memory overhead while retaining the benefits of end-to-end optimization. We further introduce a lightweight variant of ZeroQAT for quantized fine-tuning, which freezes and pre-quantizes most parameters to further cut memory usage. Experiments show that ZeroQAT consistently outperforms representative PTQ and QAT baselines while requiring significantly less memory. For example, ZeroQAT enables fine-tuning of a 13B model at extremely low bit-widths (e.g., 2-4 bits) on a single 8GB GPU, and even allows fine-tuning a 6.7B model on a OnePlus 12 smartphone, demonstrating its practicality for end-to-end QAT on resource-limited edge devices. Our code is released at `https://anonymous.4open.science/r/ZO_quantization-2DEB`.

## 1 Introduction

Large language models (LLMs) have emerged as essential tools for advancing natural language understanding and generation, driving progress in both research and industrial applications (Yang et al., 2019; Liu et al., 2019; Talmor et al., 2018; Chowdhery et al., 2023; Zheng et al., 2020). Despite their transformative potential, training and deploying these models incur extremely high computational and memory costs. Such requirements not only constrain accessibility and scalability but also limit practicality in resource-constrained environments, including mobile and edge devices, embedded systems, and even enterprise servers with strict hardware or budget limitations (Zeng et al., 2024; Chen et al., 2024a; Tan et al., 2025).

To address these challenges, model compression has been widely studied, with quantization being one of the most effective and indispensable techniques for deployment. Quantization methods are generally divided into post-training quantization (PTQ) and quantization-aware training (QAT). PTQ is simple and widely adopted as it avoids retraining, while QAT usually achieves higher accuracy when resources permit. However, for LLMs the memory demand of QAT is prohibitive (Team et al., 2025). For example, fine-tuning LLama-7B may require hundreds of gigabytes of GPU memory, and larger models often need multi-node clusters, which severely limits practicality. As a result, PTQ dominates in practice, not for its superiority but feasibility.

In low-bit scenarios, the adaptation capability for distribution shifts and mitigate performance degradation becomes the key factor that determines whether a quantization method can preserve model quality. This adaptation capability reflects how well the method can handle the distortions introduced by quantization, with stronger adaptation generally leading to more reliable performance. Range-based PTQ (Jacob et al., 2018; Nagel et al., 2019; Xiao et al., 2023), which derives parameters from activation or weight ranges, offers limited adaptation and often loses accuracy. More advanced PTQ

Table 1: Comparison of our method with existing methods. PEFT indicates parameter-efficient fine-tuning. WO and WA indicate weight-only and weight-activation quantization, respectively.

| Method | Category | Quant Support | Low-bit performance Pre-train | Fine-tune | Memory Efficiency |
|--------|----------|---------------|-------------------------------|-----------|-------------------|
| SmoothQuant | Range PTQ | WA | ✗ | ✗ | High |
| GPTQ | Approx PTQ | WO | ✗ | ✗ | High |
| OmniQuant | Approx PTQ | WA | ✓ | ✗ | Moderate |
| LLM-QAT | Full QAT | WA | ✓ | ✓ | Low |
| QLoRA | PEFT QAT | WO | ✓ | ✓ | Moderate |
| EfficientQAT | PEFT QAT | WO | ✓ | ✓ | High |
| ZeroQAT | Full/PEFT QAT | WA | ✓ | ✓ | High |

methods, such as approximation-based approaches (Nagel et al., 2020; Li et al., 2021; Frantar et al., 2022; Shao et al., 2023), better align with full-precision outputs but are still not end-to-end optimization schemes. As a result, they often introduce two characteristic issues: cumulative errors and objective inconsistency, hinder accuracy especially in low-bit settings. These issues are amplified in fine-tuned models, which are highly task-specific and sensitive to quantization perturbations (Dong et al., 2021). Consequently, PTQ often delivers unsatisfactory accuracy in deployment.

QAT provides a principled solution by modeling quantization effects during training, allowing the model to mitigate quantization errors. While QAT shows strong robustness in low-bit regimes (below 8 bits), its prohibitive memory footprint from backpropagation limits applicability to large-scale models. Recent advances in zeroth-order (ZO) optimization, which estimate gradients using only forward passes (e.g., finite differences), significantly reduce memory usage by avoiding storage of activations and optimizer states, offering a promising path for memory-efficient fine-tuning. This naturally raises the question: *Can ZO be combined with QAT to achieve high-quality low-bit quantization of LLMs, with memory efficiency comparable to inference?*

In this work, we propose ZeroQAT, the first end-to-end QAT framework supporting both low-bit weight and activation on-device quantization. As shown in Table 1, ZeroQAT reduces the resource burden of conventional QAT while mitigating the accuracy loss commonly seen in PTQ. Unlike prior methods that require massive computing resources (Liu et al., 2023), ZeroQAT updates model parameters using gradients estimated purely from forward passes, reducing memory usage to inference-level and making QAT feasible even on edge devices. It further integrates learnable weight clipping and activation transformations, optimized jointly with model parameters via ZO. Moreover, a lightweight variant is devised for further memory reduction. Experiments on both quantized pre-training and fine-tuning show that ZeroQAT consistently outperforms representative PTQ and QAT baselines. For instance, it improves accuracy by 5.1% on average over five zero-shot tasks and 9.1% on four downstream tasks under 2-bit weight-only quantization. More importantly, ZeroQAT overcomes the memory barrier of QAT, enabling training of 13B LLM on a single 8GB low-end GPU and even fine-tuning 6.7B model on OnePlus 12 smartphone. This capability makes end-to-end on-device QAT practical on resource-constrained edge devices.

In summary, our major contributions are as follows: 1) We conduct a preliminary study of PTQ and QAT in low-bit pre-training and fine-tuning, revealing their weaknesses and causes of performance degradation. 2) We propose ZeroQAT, a novel end-to-end zeroth-order QAT framework that achieves high-quality low-bit quantization with inference-level cost. 3) We conduct extensive evaluation across LLM architectures, datasets, and quantization settings, showing consistent accuracy and memory improvements over prior PTQ and QAT baselines. 4) We validate on mobile devices, where ZeroQAT can fine-tune OPT-6.7B on OnePlus 12 smartphone while full-precision ZO fine-tuning is infeasible, demonstrating its practicality for real-world deployment.

## 2 BACKGROUND AND RELATED WORKS

**Quantization.** In this work, we mainly study the widely used uniform quantization (Jacob et al., 2018) for its better efficiency. The quantization process can be formulated by:

$$\overline{\mathbf{X}}_{\text{INT}} = \text{clamp}\left(\left\lceil \frac{\mathbf{X}_{\text{FP16}}}{\Delta} \right\rfloor + z, Q_N, Q_P\right)$$

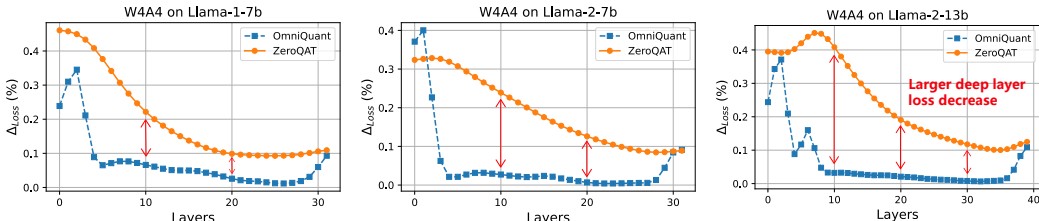

Figure 1: Comparison of the layer-wise reconstruction loss reduction between OmniQuant (Shao et al., 2023) and our method. X-axis is the index of layer, Y-axis measures the ratio of loss decrease.

where $\mathbf{X}$ is the floating-point tensor, $\overline{\mathbf{X}}$ is the quantized counterpart, $\lceil \cdot \rfloor$ is rounding operation, $N$ is the target bit number, $\Delta$ and $z$ denote the step size and zero-point offset value respectively. For symmetric quantization, $Q_N = -2^{N-1}$, $Q_P = 2^{N-1} - 1$, $\Delta = \frac{\max(|\mathbf{X}|)}{Q_P}$ and $z = 0$. Whereas for asymmetric quantization, $Q_N = 0$, $Q_P = 2^N - 1$, $\Delta = \frac{\max(|\mathbf{X}|) - \min(|\mathbf{X}|)}{Q_P}$ and $z = -\lceil \frac{\min(|\mathbf{X}|)}{\Delta} \rfloor$. In this paper, we focus on the asymmetric quantization scheme for its better accuracy.

**Layer-wise calibration.** Layer-wise calibration strategy is the most widely adopted approach in approximation-based PTQ, because it is relatively efficient in terms of memory, computation, and data usage. The key idea is to minimize quantization error via reconstruction objectives. For example, the widely used layer-wise reconstruction loss minimizes the squared error, relative to the full precision layer output (Li et al., 2021; Shao et al., 2023). Formally, when both weights and activations are quantized, this can be stated as

$$\arg\min_{\overline{W}^l} \|W^l X^l - \overline{W}^l \overline{X}^l\|_2^2. \tag{1}$$

where $\overline{W}, \overline{X}$ are the quantized version of weight and activations, $l$ indicates the $l$-th layer.

We present our related works section in Appendix B.

## 3 CHALLENGE OF EXISTING QUANTIZATION METHODS

### 3.1 CHALLENGE OF EXISTING POST TRAINING QUANTIZATION METHODS

**Range-based PTQ.** These methods rescale or clip weight and activation ranges to reduce quantization error. They are computationally efficient and perform reasonably well at moderate bit-width. For example, SmoothQuant (Xiao et al., 2023) achieves a perplexity of 6.20 in W6A6 (i.e., quantization using 6 bits weight and 6 bits activation), close to the full-precision 5.47 (Table 2). However, their limited adaptation to distributional and semantic characteristics leads to severe degradation at low bit-widths. For example, under W4A4, SmoothQuant's perplexity deteriorates to 83.12 versus 5.47 in full precision.

**Approximation-based PTQ.** These methods narrow the gap between quantized and full-precision outputs via techniques such as learned rounding or reconstruction, adapting to data distributions and model behavior. However, there are two issues still remain and are exacerbated in low-bit quantization settings.

Here, we take a representative approximation-based PTQ method, OmniQuant (Shao et al., 2023), as an example to illustrate the two issues. 1) Cumulative error propagation. To measure error propagation, we compute relative loss reduction across layers, $\Delta_{Loss} = (\mathcal{L}_{before} - \mathcal{L}_{after})/\mathcal{L}_{before}$, where $\mathcal{L}_{before}$ and $\mathcal{L}_{after}$ denote reconstruction loss before and after optimization. As shown in Figure 1, OmniQuant improves shallow layers but benefits diminish in deeper ones, since each layer is optimized on activations

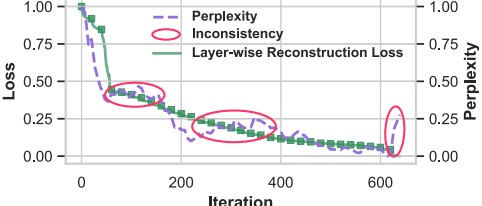

Figure 2: Objective inconsistency between the reconstruction loss used by approximation-based PTQ and final evaluation metrics.

Table 2: Results of applying different quantization methods on LLama2-7B. ‡ indicates that the method is intrinsically not suitable for the setting; we report these results to illustrate its limitations.

| Method | Category | Quantized Pre-training (PPL ↓) | | | Quantized Fine-tuning (Acc ↑) | | |
|--------|----------|------|-------|------|------|-----------|------|
| | | W6A6 | W2A16 | W4A4 | W6A6 | W2A16g128 | W4A4 |
| ZO (FP16) | - | | 5.47 | | | 66.0 | |
| Zero-shot | - | | - | | | 41.3 | |
| SmoothQuant | Range-based PTQ | 6.20 | 100.23$^{‡}$ | 83.12 | 57.2 | 27.7$^{‡}$ | 32.9$^{‡}$ |
| OmniQuant | Approx-based PTQ | 5.87 | 37.37 | 14.26 | 63.9 | 40.6$^{‡}$ | 38.8$^{‡}$ |
| EfficientQAT | QAT | 5.60 | 33.40 | 76.32$^{‡}$ | 66.4 | 45.4 | 28.6$^{‡}$ |
| ZeroQAT | QAT | 5.76 | 29.61 | 12.95 | 65.3 | 54.1 | 55.7 |

already perturbed by prior quantization noise, making it increasingly difficult to suppress the reconstruction error. This cumulative error propagation constrains overall quantization quality. 2) Objective inconsistency. OmniQuant uses layer-wise reconstruction loss (see Eq.1) as training objective, assuming lower reconstruction loss is aligned with lower perplexity and better downstream accuracy. However, as shown in Figure 2, this alignment does not always hold, in several training stages (highlighted in red), reconstruction loss decreases while perplexity fluctuates. This indicates that local layer-level improvements do not reliably translate into global task-level gains, making reconstruction loss a suboptimal proxy for end-to-end performance, especially under low-bit quantization.

**Failure on fine-tuned model.** When PTQ is applied to fine-tuned LLMs, it often fails to preserve task accuracy under low-bit settings. As shown in Table 2, SmoothQuant maintains moderate accuracy at W6A6 (57.2% vs. 66.0% in FP16) but drops to 32.9% at W4A4. Similarly, OmniQuant achieves 63.9% at W6A6, close to FP16, yet falls to 38.8% at W4A4 despite optimization-based techniques. These results indicate that while PTQ remains viable at moderate bit-widths, its effectiveness collapses under aggressive compression, in some cases nearly destroying task performance.

## 3.2 CHALLENGE OF EXISTING QUANTIZATION-AWARE TRAINING METHODS

Compared with PTQ, QAT offers stronger adaptation by compensating for quantization errors during training. However, its computational and memory costs are prohibitive for LLMs (Liu et al., 2023). To reduce this overhead, later works combine QAT with parameter-efficient methods such as LoRA (Dettmers et al., 2023; Xu et al., 2023; Li et al., 2023) or update only quantizer parameters (Chen et al., 2024b), achieving competitive results in weight-only quantization. Yet their effectiveness drops in low-bit joint weight-activation settings, as shown in Table 2, EfficientQAT maintains reasonable perplexity at W6A6 (5.60) and W2A16 (33.40), but degrades sharply at W4A4 (76.32), highlighting the difficulty of modeling dynamic activations.

Overall, although QAT methods can surpass PTQ in some settings, they have not consistently delivered strong results for both weight and activation quantization at aggressive bit-widths under realistic resource constraints. Recent efforts that combine zeroth-order (ZO) optimization with quantization primarily target weight-only scenarios (Zhou et al., 2025; Shang et al., 2025), thus leaving the challenges of low-bit activation quantization unresolved. Motivated by this gap, we develop a ZO-based QAT framework that, to the best of our knowledge, is the first to maintain superior accuracy in both low-bit weight and activation settings.

## 4 ZEROQAT

In this section, we present ZeroQAT, which enables adaptive fine-tuning of both model and quantization parameters with low memory requirements. We employ zeroth-order stochastic gradient descent to estimate gradients solely from quantized model inference, and introduce adaptive smoothing and weight quantization strategies to improve low-bit performance. Unlike prior works that rely on hand-crafted or layer-wise local objectives, ZeroQAT jointly optimizes model and quantization parameters in an end-to-end manner, yielding superior accuracy. In addition, we propose a lightweight variant to further cut memory cost during quantized fine-tuning.

## 4.1 QUANTIZATION-AWARE ZEROTH-ORDER OPTIMIZATION

Unlike conventional first-order optimization that computes gradients via backpropagation, zeroth-order (ZO) optimization estimates them using only function queries through finite differences (Chen et al., 2023; Liu et al., 2018; Ye et al., 2018). This avoids storing activations, backward gradients, and optimizer states, greatly reducing memory costs in LLM fine-tuning. For each random direction, ZO requires only two forward passes to approximate the gradient, given a mini-batch $\mathcal{B}$:

$$\hat{\nabla}\mathcal{L}(\overline{W}; \mathcal{B}) = \frac{1}{q} \sum_{i=1}^{q} \left[ \frac{\mathcal{L}\left(Q(W + \epsilon u_i); \mathcal{B}\right) - \mathcal{L}\left(Q(W - \epsilon u_i); \mathcal{B}\right)}{2\epsilon} u_i \right], \tag{2}$$

where $Q$ is the quantizer, $\overline{W}$ is the quantized parameters, $u_i \in \mathcal{N}(0, \mathbf{I})$ is a random perturbation, $q$ is the number of directions, and $\epsilon > 0$ is a small scalar.

Following QAT practice, we maintain full-precision weights while using their quantized counterparts in forward passes. Unlike FO-QAT, ZeroQAT does not require the straight-through estimator (STE) (Bengio et al., 2013), since gradients are estimated directly via zeroth-order finite differences, bypassing the non-differentiability of the quantizer. Given a learning rate $\eta$ and a mini-batch $\mathcal{B}_t$ at iteration $t$, the update rule becomes:

$$W_{t+1} = W_t - \eta\hat{\nabla}\mathcal{L}(\overline{W}_t; \mathcal{B}_t). \tag{3}$$

In ZeroQAT, the ZO estimator remains unbiased with respect to the gradient of a smoothed quantized objective, which ensures standard convergence guarantees. In contrast, QAT methods based on the STE rely on a hand-crafted surrogate gradient that introduces inherent bias. This bias becomes particularly severe in low-bit regimes, where the true smoothed gradients are already small but STE still produces large surrogate updates, leading to unstable or suboptimal convergence. A formal analysis and quantitative bounds on this bias are provided in Appendix G.

## 4.2 ADAPTIVE OUTLIER SMOOTHING AND WEIGHT QUANTIZER

**Adaptive outlier smoothing.** Due to the quantization error caused by the extreme activation outliers in specific channels, which expand the dynamic range and degrade quantization precision for normal activation values, the previous methods (Xiao et al., 2023; Wei et al., 2022; Shao et al., 2023) migrate the difficulty of activation quantization to weight quantization with a mathematically equivalent smoothing, as the weights are generally more uniform and thus easier to be quantized. However, relying on either hand-crafted smoothing parameters or layer-wise calibrated smoothing often results in suboptimal performance, due to the lack of end-to-end joint optimization.

In contrast, our QAT framework enables end-to-end joint optimization of smoothing parameters along with model parameters, thereby improving consistency and reducing quantization error. Inspired by previous works such as SmoothQuant (Xiao et al., 2023) and Outlier Suppression+ (Wei et al., 2022), which statically manipulate activation distributions via channel-wise scaling and shifting, we adapt these techniques into a jointly optimized framework to dynamically mitigate activation outliers during training, providing an effective solution for the outlier issue. Specifically, we represent the computation of a linear layer as:

$$\mathbf{Y} = \mathbf{X}\mathbf{W} + \mathbf{B} = \underbrace{[(\mathbf{X} - \delta) \oslash s]}_{\bar{\mathbf{X}}} \cdot \underbrace{[s \odot \mathbf{W}]}_{\bar{\mathbf{W}}} + \underbrace{[\mathbf{B} + \delta\mathbf{W}]}_{\bar{\mathbf{B}}} \tag{4}$$

where $\mathbf{X} \in \mathbb{R}^{T \times D_1}$, the $T$ is the sequence length, $\mathbf{W} \in \mathbb{R}^{D_1 \times D_2}$ is the weight matrix and $\mathbf{B} \in \mathbb{R}^{1 \times D_2}$ is the bias. Here, $s$ and $\delta$ are learnable channel-wise scaling and shifting parameters, jointly optimized during training, $\bar{\mathbf{X}}, \bar{\mathbf{W}}$ and $\bar{\mathbf{B}}$ represent the smoothed activation, weight and bias, respectively, $\oslash$ and $\odot$ are element-wise division and multiplication.

**Adaptive weight quantizer.** As demonstrated by previous work, some weights play a significant role in the performance of the model, naive uniform quantization can cause significant performance degradation. Similar to previous QAT methods that adopt learnable step size and zero-point parameters (Esser et al., 2019; Bhalgat et al., 2020), we also conduct weight quantization with the learnable step size and offset. However, due to the activation-weight smoothing introduced in our framework, the weight distributions in some channels become skewed, resembling the activation distributions

and deviating from the typically assumed uniformity. Therefore, we jointly learn clipping thresholds to adaptively determine the optimal clipping range for weights.

Specifically, considering asymmetric quantization, the quantization of weights as formulated by

$$\overline{W} = \text{clamp}(\lceil \frac{W}{\Delta} \rfloor + z, \alpha \cdot Q_P, \beta \cdot Q_P) \tag{5}$$

where $\Delta$ and $z$ are learnable step size and zero-point, respectively, initialized based on the default asymmetric quantization scheme. $\alpha$ and $\beta$ are learnable clipping coefficients (with $\alpha < \beta$), and $Q_P$ denotes the maximum positive quantization level. Intuitively, for weights with near-uniform distributions after smoothing, $\alpha$ and $\beta$ converge to similar values, resulting in a tight clipping range that preserves precision. In contrast, for biased weight distributions, $\alpha$ and $\beta$ adapt to asymmetrically clip the dynamic range, thereby mitigating the impact of outliers.

### 4.3 LIGHTWEIGHT ZEROQAT FOR MEMORY REDUCTION IN QUANTIZED FINE-TUNING

We further propose a lightweight variant of ZeroQAT designed specifically for quantized fine-tuning, to substantially reduce the fine-tuning memory footprint. It is worth noting that this strategy is effective only in fine-tuning, applying it to quantized pre-training leads to noticeable performance degradation (see Appendix D.3).

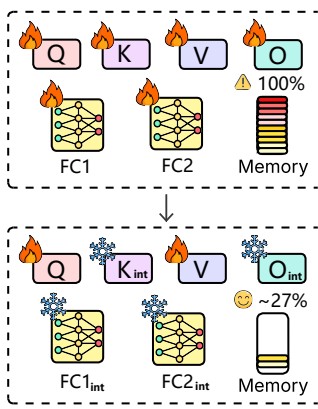

Unlike backpropagation-based methods, where memory is dominated by weights, activations, and optimizer states, ZeroQAT's cost mainly comes from the parameters actively updated during fine-tuning. Pre-quantizing the entire model could further reduce memory, but this fails in practice. As small ZO perturbations are rounded away while large ones destabilize training, making naive full-model pre-quantization unsuitable.

To overcome this, we introduce a lightweight variant. Most parameters are frozen and pre-quantized, while only the query (Q) and value (V) matrices of attention layers are kept in full precision, as illustrated in Figure 3. Thus, memory use comes from the full-precision Q and V plus quantized frozen weights. This design

Figure 3: lightweight Zero-QAT for fine-tuning.

substantially reduces the fine-tuning footprint while retaining sufficient trainable capacity for adaptation. This enables fine-tuning large models such as OPT-13B under low-bit settings with memory as low as 6.8 GB (in Table 7), far lighter than existing QAT baselines.

## 5 EXPERIMENT

We present a comprehensive evaluation of ZeroQAT, reporting results on both quantized pre-training and quantized fine-tuning (Sections 5.1 and 5.2), followed by ablation studies to assess the contributions of different design (Section 5.3). We then provide an efficiency analysis including memory and speed (Section 5.4). Hyperparameter settings are detailed in Appendix C.1. GPU-end experiments are conducted on an NVIDIA A100, and device-end experiments are conducted on a OnePlus 12 smartphone with a Snapdragon 8 Gen 3 SoC and 16GB RAM. All results are averaged over three runs.

### 5.1 ZEROQAT FOR QUANTIZED PRE-TRAINING

**Training and evaluation.** For the parameters of smoothing and weight clipping, we leverage reconstruction loss for a lightweight initialization, and then jointly train with the model via ZO. For LLama-series weight-only quantization, we retain only weight clipping. Pre-training uses mixed segments from WikiText2 and C4, with perplexity measured on three pretraining context datasets. We further evaluate zero-shot accuracy on five datasets under GPTQ settings with lm-eval-harness. More details including baselines are provided in Appendix C.2.

**Perplexity Results.** We target to examine the intrinsic language modeling performance of the quantized model. The perplexity results of LLama-series and OPT-series models are presented in Table 3

Table 3: Weight-only and weight-activation quantization results of Llama-series models on two datasets: WikiText2 (WIKI), and C4. The results on OPT models is reported in Table E.1.

| Llama / PPL ↓ | | Llama1-7B | | Llama1-13B | | Llama2-7B | | Llama2-13B | |
| Task | | WIKI | C4 | WIKI | C4 | WIKI | C4 | WIKI | C4 |
|---|---|---|---|---|---|---|---|---|---|
| FP16 | - | 5.68 | 7.08 | 5.09 | 6.61 | 5.47 | 6.97 | 4.88 | 6.46 |
| W2A16 | RTN | 1.1e5 | 1.3e5 | 6.8e4 | 5.6e4 | 3.8e4 | 4.8e4 | 5.6e4 | 7.2e4 |
| | GPTQ | 5.6e4 | 689.13 | 5.5e3 | 6.97 | 7.7e3 | NAN | 2.1e3 | 323.12 |
| | OmniQuant | 15.47 | 24.89 | 13.21 | 18.31 | 37.37 | 90.64 | 17.21 | 26.76 |
| | ZeroQAT | **12.85** | **17.47** | **10.29** | **15.37** | **29.61** | **55.34** | **15.97** | **24.68** |
| W6A6 | SmoothQuant | 6.03 | 7.47 | 5.42 | 6.97 | 6.20 | 7.76 | 5.18 | 6.76 |
| | OmniQuant | 5.96 | 7.43 | **5.28** | **6.84** | 5.87 | **7.48** | 5.14 | 6.74 |
| | ZeroQAT | **5.85** | **7.47** | 5.96 | 7.01 | **5.76** | 8.81 | **5.10** | **6.70** |
| W4A4 | SmoothQuant | 25.25 | 32.32 | 40.05 | 47.18 | 83.12 | 77.27 | 35.88 | 43.19 |
| | OmniQuant | 11.26 | **14.51** | 10.87 | 13.78 | 14.26 | 18.02 | 12.30 | 14.55 |
| | ZeroQAT | **11.10** | 14.78 | **10.04** | **12.65** | **12.95** | **16.73** | 10.41 | **12.43** |

and Table E.1 respectively. Under the rather easier quantization setting W6A6, the baselines and our method achieve similar, almost lossless performance compared with full precision, absolute perplexity gap is smaller than one. More importantly, under the hard quantization setting W2A16(g128) and W4A4, because our method has better adaptation capability by enabling fine-tuning of the whole model, one can see that ZeroQAT consistently outperforms the baseline methods, yielding lower perplexity across both model families and datasets. This highlights the effectiveness of ZeroQAT in preserving model quality under aggressive quantization.

Table 4: Weight-only and weight-activation quantization results of LLama models. This table reports the accuracy of 5 zero-shot tasks. Results of Llama-1-13B are shown in Table E.2.

| Llama / Acc ↑ | #Bits | Method | PIQA | ARC-e | ARC-c | HellaSwag | Winogrande | Avg. |
|---|---|---|---|---|---|---|---|---|
| | FP16 | - | 77.47 | 72.38 | 41.46 | 73.00 | 67.07 | 65.26 |
| | W2A16 | RTN | 47.33 | 28.17 | 25.17 | 25.10 | 47.50 | 34.67 |
| | W2A16 | GPTQ | 57.38 | 36.62 | 25.00 | 42.50 | 49.38 | 40.35 |
| | W2A16 | EfficientQAT | 62.25 | 48.12 | 27.75 | 47.50 | 53.37 | 47.65 |
| | W2A16 | ZeroQAT | 68.25 | 53.87 | 27.62 | 51.62 | 57.38 | **51.75** |
| Llama-1-7B | W4A4 | SmoothQuant | 49.80 | 30.40 | 25.80 | 27.40 | 48.00 | 38.41 |
| | W4A4 | LLM-QAT | 51.50 | 32.57 | 28.63 | 31.10 | 51.90 | 41.39 |
| | W4A4 | LLM-QAT+SQ | 55.93 | 35.90 | 30.60 | 44.80 | 50.60 | 46.72 |
| | W4A4 | OS+ | 62.70 | 39.20 | 32.64 | 47.89 | 52.96 | 49.60 |
| | W4A4 | OmniQuant | 67.38 | 53.87 | 30.63 | 53.12 | 55.25 | 52.15 |
| | W4A4 | ZeroQAT | 66.98 | 54.12 | 32.19 | 57.85 | 54.37 | **53.11** |

**Zero-shot Accuracy Results.** Moreover, Table 4 reports the zero-shot results of LLama-7B on five downstream datasets evaluated by accuracy. As expected, the FP16 setting achieves the highest average accuracy, serving as the upper bound. Under both the W2A16 and W4A4 configurations, ZeroQAT consistently outperforms other quantization approaches, yielding higher average accuracy across both model scales, for instance, significantly increasing 5.1% accuracy in 2-bit weight-only quantization. This result demonstrates that ZeroQAT maintains strong task generalization even when quantization is pushed to low-bit precision.

## 5.2 ZeroQAT for Quantized Fine-tuning

**Training and Evaluation.** Following prior work, we fine-tune models on a small subset of Alpaca and evaluate across multiple benchmarks, including commonsense reasoning, classification, and question answering tasks. We adopt a few-shot fine-tuning protocol with fixed quantization parameters and report averaged results over three runs. Full experimental details and baselines are provided in Appendix C.3.

Table 5: Experimental results of quantized fine-tuning on OPT models.

| OPT / Acc ↑ | | OPT-2.7B | | | | OPT-6.7B | | | | OPT-13B | | | |
|---|---|---|---|---|---|---|---|---|---|---|---|---|---|
| Task | | SST-2 | CB | SQuAD | DROP | SST-2 | CB | SQuAD | DROP | SST-2 | CB | SQuAD | DROP |
| Zero-shot | - | 56.3 | 50.0 | 29.8 | 10.0 | 64.2 | 50.0 | 37.9 | 13.1 | 58.8 | 46.4 | 46.2 | 14.6 |
| FP16 (ZO) | | 90.0 | 69.6 | 68.7 | 22.9 | 90.2 | 71.4 | 76.0 | 26.4 | 91.4 | 67.9 | 84.7 | 30.9 |
| | RTN | 44.4 | 44.6 | 0.0 | 0.0 | 59.2 | 50.0 | 0.0 | 0.0 | 53.5 | 50.0 | 0 | 0 |
| | QLoRA | 61.2 | 51.8 | 0.0 | 8.2 | 64.8 | 58.9 | 0.0 | 0.0 | 63.8 | 69.6 | 0 | 0 |
| W2A16g128 | OmniQuant | 72.8 | 55.4 | 16.5 | 4.4 | 61.6 | 55.3 | 27.7 | 12.6 | 62.6 | 29.8 | 38.8 | 16.4 |
| | EfficientQAT | 76.6 | 57.1 | 29.0 | 12.6 | 75.6 | 58.9 | 32.4 | 14.6 | 81.2 | 62.5 | 46.7 | 16.9 |
| | ZeroQAT | **85.2** | **62.5** | **36.9** | **16.6** | **84.8** | **67.8** | **46.7** | **18.9** | **85.6** | **64.2** | **59.6** | **22.9** |
| | SmoothQuant | 56.0 | 55.4 | 7.6 | 5.4 | 58.8 | 50.0 | 12.8 | 6.2 | 57.5 | 52.4 | 13.4 | 7.1 |
| W4A4 | OmniQuant | 59.2 | 60.7 | 22.1 | 6.7 | 61.2 | 48.2 | 24.7 | 11.7 | 59.2 | 50.0 | 28.8 | 13.5 |
| | ZeroQAT | **87.8** | **66.1** | **47.8** | **13.3** | **87.9** | **64.3** | **51.1** | **19.3** | **88.2** | **62.1** | **62.4** | **24.3** |

**Results**. We evaluate quantized fine-tuning on OPT models (2.7B, 6.7B, and 13B) across two classification tasks (SST-2, CB) and two QA generation tasks (SQuAD, DROP). For PTQ methods such as SmoothQuant and OmniQuant, we first fine-tune the models in full precision using ZO to ensure comparable starting points, and then apply the corresponding quantization method. In contrast, QAT methods, including ZeroQAT, directly produce quantized models during fine-tuning without the need for a separate PTQ stage.

The results are summarized in Table 5. Fine-tuning adapts model parameters to narrow task-specific optima (Dong et al., 2021), which increases their sensitivity to quantization noise. Consequently, less adaptive PTQ methods suffer from severe degradation in low-bit settings. By comparison, ZeroQAT consistently delivers higher accuracy across all tasks and model scales, in some cases approaching FP16 performance. For example, under the W4A4 setting, ZeroQAT achieves about 88% accuracy on SST-2 across the three OPT models, whereas baseline methods remain around 60%. We also fine-tuned LLama-1 models on Alpaca, with results shown in Table 6. ZeroQAT again outperforms prior methods across different bit-widths and model sizes. For instance, when quantizing LLama-7B and LLama-13B weights to 2 bits, ZeroQAT achieves absolute accuracy improvements of 4.8% and 3.6% over the best baseline EfficientQAT, illustrating the effectiveness of our approach.

Table 6: Averaged accuracy over 5 datasets after fine-tuning. Evaluation on MMLU is presented in Appendix F

| Method | #Bits | 7B | 13B |
|---|---|---|---|
| - | FP | 67.0 | 69.3 |
| QLoRA w/GPTQ | W2A16 | 31.8 | 32.4 |
| QA-LoRA | W2A16 | 34.6 | 37.3 |
| IR-QLoRA | W2A16 | 34.4 | 36.3 |
| PEQA | W2A16 | 35.2 | 34.8 |
| EfficientQAT | W2A16 | 49.1 | 52.1 |
| ZeroQAT | W2A16 | **53.9** | **55.7** |
| SmoothQuant | W4A4 | 37.4 | 41.6 |
| OmniQuant | W4A4 | 52.3 | 54.2 |
| ZeroQAT | W4A4 | **54.8** | **57.4** |

## 5.3 ABLATION STUDY

In this section, we conduct ablation study to examine the effectiveness of the strategies adopted in our method. More experiments are shown in Appendix D.

**Effect of initialization for Smoothing Parameters.** We initialize the smoothing parameters by minimizing reconstruction loss before applying ZO, to examine the impact of initialization quality, we conduct an ablation study by varying the number of initialization epochs, as reported in Table D.5. The results show that initialization has a clear effect on performance. With 0 epochs of initialization, performance drops noticeably across different models, while additional epochs (e.g., 20) can further improve accuracy. However, considering both performance gains and computational cost, we adopt two epochs as the default initialization setting.

## 5.4 EFFICIENCY OF ZEROQAT

To highlight the advantage that our method enables generating a quantized and fine-tuned model in a lightweight end-to-end pipeline, we evaluate the efficiency of ZeroQAT on both a GPU server and a mobile device to demonstrate its practicality across deployment scenarios.

Table 7: Memory consumption and wallclock time per update during quantized pre-training under the W2A16g128 setting. Since ZeroQAT only stores the weights in memory, its memory usage remains unaffected by batch size.

| Method | OPT-1.3B | | OPT-2.7B | | OPT-6.7B | | OPT-13B | |
|---|---|---|---|---|---|---|---|---|
| | Memory | Time | Memory | Time | Memory | Time | Memory | Time |
| Quantized Pre-training (avg sequence length = 2048) | | | | | | | | |
| LLM-QAT (bsz=1) | 28.8GB | 1.00s | 58.6GB | 1.64s | ∼166GB | ∼5.0s | ∼337GB | ∼15.5s |
| OmniQuant (bsz=1) | 6.1GB | 0.92s | 7.4GB | 1.49s | 12.3GB | 2.65s | 16.8GB | 4.77s |
| ZeroQAT (bsz=1) | 3.1GB | 0.58s | 6.1GB | 0.98s | 14.2GB | 1.77s | 26.6GB | 3.12s |
| OmniQuant (bsz=4) | 14.7GB | 2.55s | 16.2GB | 4.03s | 22.5GB | 6.35s | 28.5GB | 11.81s |
| ZeroQAT (bsz=4) | 3.1GB | 1.72s | 6.1GB | 2.76s | 14.2GB | 4.48s | 26.6GB | 7.74s |
| Quantized Fine-tuning (max sequence length = 384) | | | | | | | | |
| EfficientQAT (bsz=1) | 2.1GB | 0.13s | 3.1GB | 0.21s | 4.4GB | 0.36s | 7.3GB | 0.67s |
| ZeroQAT (bsz=1) | 0.8GB | 0.04s | 1.5GB | 0.07s | 3.7GB | 0.18s | 6.8GB | 0.32s |
| EfficientQAT (bsz=16) | 5.9GB | 0.69s | 8.1GB | 1.10s | 11.9GB | 1.70s | 17.2GB | 3.26s |
| ZeroQAT (bsz=16) | 0.8GB | 0.31s | 1.5GB | 0.53s | 3.7GB | 0.94s | 6.8GB | 1.73s |

**Server-side Efficiency.** Table 7 compares memory requirements and wallclock time per update across QAT and PTQ methods. For quantized pre-training, ZeroQAT reduces memory usage by 89-92% relative to the costly LLM-QAT, while also accelerating training. Compared to the PTQ method OmniQuant, ZeroQAT offers clear advantages, for instance, it halves memory use (OPT-1.3B: 6.1GB to 3.1GB) and achieves about 1.5× faster updates (OPT-2.7B: 1.49s to 0.98s). For quantized fine-tuning, ZeroQAT's memory-efficient design requires storing only weights, making usage independent of batch size. Against EfficientQAT, it consistently saves memory and improves throughput, especially on smaller models such as OPT-1.3B, reducing memory by 86% (5.9GB to 0.8GB) and wallclock time by 55% (0.69s to 0.31s) with the same batch size.

Table 8: Evaluation of memory consumption and speed on a OnePlus 12 smartphone under W4A4 quantization. Prompts of 384 tokens are used in inference, and OOM indicates out of memory.

| Stage | Metrics | OPT-1.3B | | OPT-2.7B | | OPT-6.7B | |
|---|---|---|---|---|---|---|---|
| | | FP16 | ZeroQAT | FP16 | ZeroQAT | FP16 | ZeroQAT |
| Fine-tuning | Latency | 11.2s | 7.8s | 19.6s | 12.3s | / | 29.1s |
| | Weight memory | 2.6GB | 0.9GB | 5.4GB | 1.8GB | 13.4GB | 4.6GB |
| | Running memory | 3.5GB | 1.2GB | 8.1GB | 2.6GB | OOM | 6.4GB |
| Inference | Token / s | 10.9 | 15.4 | 7.58 | 11.0 | 3.13 | 4.76 |
| | Speed up | 1.0× | 1.41× | 1.0× | 1.45× | 1.0× | 1.52× |

**On-device Efficiency.** Table 8 compares FP16 baseline with ZeroQAT under W4A4 for OPT-1.3B, 2.7B, and 6.7B models. The results were collected on a OnePlus 12 smartphone with a Snapdragon 8 Gen 3 SoC and 16GB RAM. ZeroQAT reduces fine-tuning latency by 30% and 37% for OPT-1.3B and OPT-2.7B, respectively, while cutting running memory from 3.5GB to 1.2GB and from 8.1GB to 2.6GB. For OPT-6.7B, FP16 fine-tuning is infeasible (OOM), whereas ZeroQAT runs within 6.4GB memory with 29.1s latency. During inference, ZeroQAT further achieves 1.41×-1.52× higher token throughput, demonstrating its practicality on resource-constrained devices.

# 6 CONCLUSION

In this paper, we proposed ZeroQAT, a zeroth-order-based quantization-aware training framework supporting both weight and activation quantization under extremely low bit-widths. We further introduced adaptive smoothing and an adaptive weight quantizer to reduce errors, and a lightweight variant that freezes and quantizes part of the model to lower fine-tuning memory cost. Experiments on quantized pre-training, fine-tuning, and on-device deployment show that ZeroQAT consistently outperforms PTQ and QAT baselines in both accuracy and efficiency, and even enables fine-tuning large LLMs on OnePlus 12 smartphone under strict memory constraints.

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

## A   CLAIM OF LLM USAGE

In this work, large language models (LLMs) were used solely as a general-purpose writing assistant. Their role was limited to correcting grammar, fixing typographical errors, and polishing the language for clarity and readability.

## B   RELATED WORK

### B.1   MODEL QUANTIZATION

Quantization techniques aim to properly map the original continuous real values to a discrete low-bit format (e.g., INT8 or INT4), leading to significant memory saving and inference acceleration while maintaining the performance (Zhou et al., 2016). Quantization techniques can be generally divided into two categories: Post-training quantization (PTQ) and quantization-aware training (QAT). The QAT method generally yields better results due to better adaptation capability, but the high retraining cost (in both memory and computation) has discouraged many researchers. Therefore, most of the LLM quantization works focus on PTQ methods, which can be mainly divided into range-based PTQ (Jacob et al., 2018; Nagel et al., 2019; Xiao et al., 2023) and approximation-based PTQ methods (Nagel et al., 2020; Li et al., 2021; Frantar et al., 2022; Shao et al., 2023). The range-based PTQ typically relies on static analysis, where the range (e.g., minimum and maximum values) of weights or activations is collected to determine quantization parameters. The approximation-based PTQ methods, with more adaptation, explicitly frame quantization as an error minimization problem, optimizing quantized parameters to closely approximate the full-precision model outputs.

### B.2   ZEROTH-ORDER OPTIMIZATION

Zeroth-order optimization (ZO), which estimates gradients using only function evaluations, has emerged as an attractive alternative to classical first-order (FO) methods. Compared to FO approaches, ZO eliminates the need for backpropagation, thereby simplifying implementation and significantly reducing memory consumption. This makes it appealing in scenarios such as adversarial attack and defense (Chen et al., 2017; Ye et al., 2018; Verma et al., 2023), machine learning explainability (Dhurandhar et al., 2018; 2019), reinforcement learning (Vemula et al., 2019), and on-chip training (Gu et al., 2021). Despite these successes, ZO optimization has been primarily applied to relatively small-scale problems, since its convergence is generally slower and suffers from high variance due to random search. These challenges are exacerbated in large-scale settings such as LLM fine-tuning, where dimensionality and resource constraints amplify the difficulty. To access further acceleration and compression, there are some works that focus on combining ZO with quantization (Zhou et al., 2025; Shang et al., 2025), while our method is the first to overcome the accuracy degradation in both low-bit weight and activation quantization scenarios.

## C   EXPERIMENTAL SETTINGS

**Quantization settings.** To comprehensively evaluate our method, we consider both weight-only and weight-activation quantization, as they represent distinct deployment scenarios. For weight-activation quantization, we adopt per-channel weight quantization and per-token activation quantization, following prior work (Dettmers et al., 2022; Shao et al., 2023). For weight-only quantization, we apply a group-wise strategy, where the weight matrix is partitioned into groups of a fixed size, and each group is assigned its own scale and zero point. Formally, for example, W2A16g128 refers to 2-bit weight-only quantization with 128 as the group size. When $g$ is omitted (e.g., W2A16), the default group size is set to the number of channels, corresponding to per-channel quantization.

### C.1   HYPERPARAMETER SETTING

We use the hyperparameters in Table C.1 for experiments on quantized pre-training and quantized fine-tuning. Specifically, pre-training prefers smaller learning rate and smaller perturbation for stable convergence, while for fine-tuning, we can use more aggressive optimization. Moreover, larger

Table C.1: The hyperparameter for experiments. For DiZO and DiZO LoRA, we only show the setting of extra hyperparameters, and have the same setting in other common hyperparameters with MeZO and MeZO LoRA respectively.

| Experiment | Hyperparameters | Values |
|---|---|---|
| Quantized Pre-training | Batch size | 4 |
| | Iteration | 10K |
| | Learning rate | {5e-7, 1e-8} |
| | Lr for smothing | 5e-6 |
| | Lr for clipping | 1e-5 |
| | Lr schedule | Linear Decay |
| | $\epsilon$ in ZO | {1e-3, 5e-4 1e-4} |
| Quantized Fine-tuning | Batch size | {32, 16} |
| | Iteration | 8K |
| | Learning rate | {1e-6, 5e-7} |
| | Lr schedule | Constant |
| | $\epsilon$ in ZO | 1e-3 |

models prefers smaller learning rate and smaller perturbation, while smaller models tend to have the opposite.

## C.2 SETTINGS OF QUANTIZED PRE-TRAINING

**Training and evaluation** Zeroth-order optimization has been shown to benefit from strong initialization (Malladi et al., 2023). To provide a stable starting point, we adopt a lightweight initialization strategy based on channel-wise scaling and shifting. Specifically, we pre-train quantized models with OmniQuant (Shao et al., 2023) for a few epochs (2 epochs in the W4A4 setting and 4 epochs in the W2A16 setting), which corresponds to roughly 10% of the full OmniQuant training cost. This initialization enables ZO to more effectively refine the quantization scales and shift factors. But for LLama-series weight-only quantization, we remove the smoothing scalar and only maintain weight clipping as smoothing only provides limited improvement. For quantized pre-training, we randomly select token segments with length 2048 and than calculate perplexity over WikiText2 (Merity et al., 2016), PTB (Marcus et al., 1994), and C4 (Raffel et al., 2020). To avoid overfitting on one specific dataset, half segments samples from WikiText2 and half from C4, while the total data size is keep same with previous work (Shao et al., 2023; Dettmers et al., 2022) and set as 128. We further assess zero-shot accuracy on a range of tasks including PIQA (Bisk et al., 2020), ARC (Clark et al., 2018), HellaSwag (Zellers et al., 2019), Winogrande (Sakaguchi et al., 2021). We adhere to the GPTQ (Frantar et al., 2022) settings for language generation experiments, and leverage the lm-eval-harness (Gao et al., 2024) tool for the evaluation of all zero-shot tasks.

**Baselines** We mainly compared with post-training quantization methods. For weight-only quantization, we compare with the vanilla round-to-nearest (RTN), GPTQ (Frantar et al., 2022). For weight-activation quantization, we compare our method with SmoothQuant (Xiao et al., 2023), RPTQ (Yuan et al., 2023), OutlierSupression+ (OS+) (Wei et al., 2022), OmniQuant (Shao et al., 2023), and one QAT method LLM-QAT (Liu et al., 2023). We keep the quantization setting of SmoothQuant and Outlier Suppression+ with per-channel weight quantization and per-token activation quantization for fair comparisons.

## C.3 SETTINGS OF QUANTIZED FINE-TUNING

**Training and evaluation.** Following existing works (Chen et al., 2024b), we fine-tune models on a small subset of Alpaca dataset (Taori et al., 2023), and report the average accuracy on datasets including PIQA, ARC, HellaSwag and Winogrande. Moreover, we fine-tune and evaluate on two classification datasets, SST-2 (Socher et al., 2013) and CB (De Marneffe et al., 2019), and two question answering datasets, SQuAD (Rajpurkar et al., 2016) and DROP (Dua et al., 2019). For these tasks, we randomly sample 1,000 examples for training, 500 for validation, and 1,000 for testing, following the common few-shot fine-tuning protocol (Malladi et al., 2023). Performance is measured using accuracy for classification tasks and F1 scores for question answering tasks. The

initialization of quantization parameters is identical to that used in quantized pre-training, and these parameters are frozen during fine-tuning. This design allows us to directly perform quantized fine-tuning without an additional quantized pre-training stage. For all fine-tuning experiments, we run our experiments three times with different seeds and report the averaged results.

**Baselines.** Beside the baseline methods used in quantized pre-training (in Section 5.1), we additionally compare our method with several leading QAT methods, including QLoRA (Dettmers et al., 2023), QA-LoRA (Xu et al., 2023), PEQA (Kim et al., 2023), IR-QLoRA (Qin et al., 2024), and EfficientQAT (Chen et al., 2024b).

# D    MORE ABLATION STUDY ON ZEROQAT

In this section, we conduct comprehensive ablation study on ZeroQAT to illustrate the effectiveness of the components or strategies we used. Specifically, the results include:

- Effect of learnable outlier smoothing and weight clipping (Table D.1).
- Effect of using fine-tuned checkpoint by first-order as PTQ's starting point (Table D.2).
- Effect of using lightweight ZeroQAT for quantized pre-training (Table D.3).
- Effect of the layer selection in lightweight ZeroQAT (Table D.4).
- Effect of quantize parameter initialization and number of training samples (Table D.5 and Table D.6).

## D.1    EFFECT OF LEARNABLE OUTLIER SMOOTHING AND WEIGHT CLIPPING

In ZeroQAT, we introduce learnable smoothing scalar and weight clipping threshold to effectively relieve the outlier issue in low-bit quantization. We conduct experiments to ablate the effectiveness of these two learnable components. As shown in Table D.1, both components positively influence performance, but learnable smoothing proves essential for weight activation quantization. Disabling it for W4A4 results in a marked increase in perplexity, mainly due to challenges with activation quantization outliers. For weight-only quantization, smoothing only offer slight improvement for less outlier occurs (Shao et al., 2023), therefore the smoothing is not used for weight-only quantization.

Table D.1: Effect of each component. WikiText2 perplexity is reported in this table. W/O indicates removing the corresponding learnable components.

| PPL ↓ | Llama-7B | | Llama2-13B | |
| --- | --- | --- | --- | --- |
| **Leanable Components** | **W4A4** | **W2A16** | **W4A4** | **W2A16** |
| Smoothing + Clipping | 12.95 | 29.32 | 10.41 | 16.04 |
| W/O Smoothing | 1.4e3 | 29.61 | 5.2e3 | 15.97 |
| W/O Clipping | 16.64 | 9.4e3 | 18.7 | 2.8e3 |
| W/O Smoothing & Clipping | 2.1e3 | 1.2e4 | 1.7e4 | 4.6e3 |

## D.2    EFFECT OF USING FIRST-ORDER FINE-TUNED MODEL FOR PTQ

When comparing our method with PTQ methods, the starting points is the full-precision fine-tuned model using ZO, therefore we investigate if the PTQ method can perform better when using fine-tuned model by first-order (FO) optimization. As shown in Table D.2, when using first-order fine-tuned model as starting point, the memory cost of fine-tuning will dramatically increase to around 100 GB, while also not enhance the performance of PTQ, yielding much lower accuracy compared with FP ZO and ZeroQAT.

## D.3    EFFECT OF USING LIGHTWEIGHT ZEROQAT FOR QUANTIZED PRE-TRAINING

In ZeroQAT fine-tuning, we devise a lightweight variant that keeps the query and value matrices in full precision while freezing and quantizing the remaining parameters. This design substantially

Table D.2: Compare with PTQ method with different fine-tuned model as starting points. Results of fine-tuning OPT-6.7B under W4A4 setting. ZO and FO indicates the starting fine-tuned checkpoint is from first-order and zeroth-order optimization respectively.

| Method | Fine-tuning Memory | PTQ memory | SST-2 | CB | SQuAD | DROP |
|---|---|---|---|---|---|---|
| FP ZO | 14.2 GB | - | 90.2 | 71.4 | 76.0 | 26.4 |
| OmniQuant (ZO) | 14.2 GB | 4.4 GB | 61.2 | 48.2 | 24.7 | 11.7 |
| OmniQuant (FO) | 98.6 GB | 4.4 GB | 58.7 | 55.3 | 31.8 | 13.5 |
| ZeroQAT | 3.7 GB | - | 87.9 | 64.3 | 51.1 | 19.3 |

reduces memory cost without sacrificing downstream task accuracy. However, when we apply the same strategy in quantized pre-training, we observe a clear performance drop, as shown in Table D.3. For example, on WikiText2, lightweight ZeroQAT yields perplexity of 41.05 and 21.97 for LLama2-7B and Llama2-13B under W2A16, compared to 29.61 and 15.95 without lightweight strategy.

This degradation can be attributed to the different optimization dynamics in pre-training versus fine-tuning. Pre-training requires updating a much larger parameter space to capture broad linguistic patterns. Freezing most of the model limits the ability to adapt quantization parameters and compensate for quantization noise, leading to accumulated errors and higher perplexity. In contrast, fine-tuning operates on narrower task-specific distributions, where updating Q and V alone is sufficient to preserve performance. These results highlight that while selective fine-tuning is effective for downstream adaptation, full-parameter optimization remains crucial in the pre-training stage under quantization.

Table D.3: Effect of using lightweight ZeroQAT in quantized pre-training. LW indicates lightweight. Perplexity on Wikitext2 is reported.

| PPL ↓ | LLama2-7b | | LLama2-13b | |
|---|---|---|---|---|
| Method | W2A16 | W4A4 | W2A16 | W4A4 |
| ZeroQAT (LW) | 41.05 | 19.34 | 21.97 | 15.45 |
| ZeroQAT | 29.61 | 12.95 | 15.95 | 10.41 |

### D.4 EFFECT OF FINE-TUNING LAYER SELECTION.

We propose a lightweight variant ZeroQAT that fine-tunes only the query (Q) and value (V) matrices in the attention layers, while freezing and quantizing the remaining parts of the model to reduce memory overhead. To evaluate the effectiveness of this strategy, we compare it with different layer selection strategy, and the results are reported in Table D.4. The results show that this selective fine-tuning approach achieves a favorable trade-off between performance and memory efficiency: it maintains accuracy comparable to full-parameter fine-tuning, while reducing memory usage to 27%-38% of the full-parameter baseline, depending on the model size. This demonstrates that restricting updates to Q and V matrices provides substantial efficiency gains without significant loss of performance.

### D.5 EFFECT OF TRAINING SAMPLE SIZE

Conventional first-order QAT methods are generally data-inefficient, as they rely on large training datasets to provide stable and accurate gradients. To examine whether ZeroQAT exhibits similar behavior, we vary the number of training samples and report the results in Table D.6. Compared to the default setting of 128 samples, changing the sample size has only a minor effect on performance, with most perplexity variations remaining within 0.5. This indicates that, unlike conventional methods, ZeroQAT does not heavily rely on large-scale data for convergence. Instead, since its gradients are estimated through noisy zeroth-order approximations, ZeroQAT benefits more from additional optimization iterations rather than larger datasets.

Table D.4: Ablation study for selecting which layers to maintain full-precision and update in Quantized Fine-tuning. The highlighted line with a blue rectangle is the setting used in ZeroQAT. Attn_Q: attention Query layer; Attn_V: attention Value layer; Attn_K: attention Key layer; Attn_O: attention output projection; Dense: dense fully connected layer.

| Attn_Q | Attn_V | Attn_K | Attn_O | Dense | W2A16g128 | | W4A4 | |
|--------|--------|--------|--------|-------|------|--------|------|--------|
| | | | | | Acc. | Memory | Acc. | Memory |
| ✓ | ✓ | ✓ | ✓ | ✓ | 55.0 | 100% | 56.8 | 91.7 |
| ✓ | ✓ | ✓ | ✓ | ✗ | 54.1 | 42% | 54.5 | 50% |
| ✓ | ✓ | ✓ | ✗ | ✗ | 54.3 | 34% | 55.4 | 44% |
| ✓ | ✓ | ✗ | ✗ | ✗ | 54.5 | 27% | 55.6 | 38% |
| ✓ | ✗ | ✗ | ✗ | ✗ | 44.3 | 20% | 46.9 | 32% |

Table D.5: Effect of the number of epochs to initialize the smoothing parameter using reconstruction loss. Perplexity on WikiText2 is reported. * indicates default setting.

| Epochs | LLama1-7B | LLama2-7B | OPT-6.7B |
|--------|-----------|-----------|----------|
| 0 | 14.33 | 15.67 | 15.49 |
| 1 | 11.68 | 13.87 | 12.53 |
| 2* | 11.10 | 12.95 | 11.48 |
| 10 | 10.86 | 12.38 | 11.12 |
| 20 | 10.20 | 12.08 | 10.95 |

Table D.6: Effect of using different number of training samples (token segments) for training.

| Samples | W2A16 | W4A4 |
|---------|-------|------|
| 32 | 30.18 | 13.32 |
| 64 | 29.87 | 13.05 |
| 128* | 29.61 | 12.95 |
| 256 | 29.65 | 12.81 |
| 512 | 29.34 | 13.06 |

# E  MORE QUANTIZED PRE-TRAINING RESULTS

To illustrate the generalizability of our method, we conduct quantized pre-training on OPT family models, and the results are shown in Table E.1. For W6A6 quantization, similar to other baselines, ZeroQAT also achieves almost loss-less results on three datasets. For more challenge W4A4 setting, ZeroQAT consistently outperforms other baselines for better adaptation.

We conduct experiment on LLama with 13B parameters, results on 5 zero-shot datasets is show in Table E.2.

Table E.1: Weight-activation quantization results of OPT models on three datasets: WikiText2 (WIKI), Penn Treebank (PT), and C4. RPTQ* represents a variant that quantizes all activations except the softmax output.

| OPT / PPL ↓ | | OPT-2.7B | | | OPT-6.7B | | | OPT-13B | | |
|-------------|-------------|-------|--------|--------|-------|-------|-------|-------|-------|-------|
| Task | | WIKI | PT | C4 | WIKI | PT | C4 | WIKI | PT | C4 |
| FP16 | - | 12.47 | 15.13 | 13.16 | 10.86 | 13.09 | 11.74 | 10.13 | 12.34 | 11.20 |
| W6A6 | SmoothQuant | 12.64 | 15.91 | 13.34 | 11.34 | 13.82 | 12.14 | 10.56 | 12.76 | 11.40 |
| | RPTQ | 13.19 | 16.37 | 14.04 | 11.19 | 13.98 | 12.08 | 11.19 | 13.98 | 12.08 |
| | RPTQ* | 12.71 | 15.53 | 13.33 | 10.96 | 13.24 | 11.86 | 10.96 | 13.24 | 11.86 |
| | OmniQuant | 12.62 | **15.32** | **13.29** | 10.96 | **13.20** | 11.81 | 10.21 | **12.47** | **11.17** |
| | ZeroQAT | **12.62** | 15.37 | 13.77 | **10.14** | 13.41 | **11.44** | 9.60 | 12.59 | 11.47 |
| W4A4 | SmoothQuant | 131.47 | 107.10 | 120.57 | 1.8e4 | 1.4e4 | 1.5e4 | 7.4e3 | 6.5e3 | 5.6e3 |
| | RPTQ | 11.45 | 14.71 | 13.12 | 12.00 | 15.17 | 12.85 | 12.74 | 15.76 | 14.71 |
| | RPTQ* | 11.45 | 14.71 | 13.12 | 17.83 | 25.10 | 19.91 | 16.45 | 23.01 | 16.80 |
| | OmniQuant | 15.65 | 23.69 | 16.51 | 12.24 | 15.54 | 13.56 | 11.65 | 15.89 | 13.46 |
| | ZeroQAT | **14.42** | **21.71** | **15.14** | **11.48** | **14.84** | **13.10** | **10.65** | **15.04** | **12.62** |

Table E.2: Weight-only and weight-activation quantization results of LLama models. This table reports the accuracy of 5 zero-shot tasks.

| LLama / Acc ↑ | #Bits | Method | PIQA | ARC-e | ARC-c | HellaSwag | Winogrande | Avg. |
|---|---|---|---|---|---|---|---|---|
| | FP16 | - | 77.47 | 72.38 | 41.46 | 73.00 | 67.07 | 65.26 |
| | W2A16 | RTN | 47.33 | 28.17 | 25.17 | 25.10 | 47.50 | 34.67 |
| | W2A16 | GPTQ | 57.38 | 36.62 | 25.00 | 42.50 | 49.38 | 40.35 |
| | W2A16 | EfficientQAT | 62.25 | 48.12 | 27.75 | 47.50 | 53.37 | 47.65 |
| | W2A16 | ZeroQAT | 68.25 | 53.87 | 27.62 | 51.62 | 57.38 | **51.75** |
| LLama-1-7B | W4A4 | SmoothQuant | 49.80 | 30.40 | 25.80 | 27.40 | 48.00 | 38.41 |
| | W4A4 | LLM-QAT | 51.50 | 32.57 | 28.63 | 31.10 | 51.90 | 41.39 |
| | W4A4 | LLM-QAT+SQ | 55.93 | 35.90 | 30.60 | 44.80 | 50.60 | 46.72 |
| | W4A4 | OS+ | 62.70 | 39.20 | 32.64 | 47.89 | 52.96 | 49.60 |
| | W4A4 | OmniQuant | 67.38 | 53.87 | 30.63 | 53.12 | 55.25 | 52.15 |
| | W4A4 | ZeroQAT | 66.98 | 54.12 | 32.19 | 57.85 | 54.37 | **53.11** |
| | FP16 | - | 79.10 | 74.83 | 42.04 | 75.62 | 70.31 | 66.33 |
| | W2A16 | RTN | 54.75 | 26.25 | 27.50 | 29.75 | 47.00 | 37.05 |
| | W2A16 | GPTQ | 59.25 | 33.00 | 25.17 | 44.25 | 53.25 | 42.98 |
| | W2A16 | EfficientQAT | 68.15 | 53.08 | 29.51 | 49.26 | 54.35 | 50.87 |
| LLama-1-13B | W2A16 | ZeroQAT | 72.41 | 57.24 | 32.12 | 53.70 | 57.54 | **54.60** |
| | W4A4 | SmoothQuant | 61.04 | 38.00 | 26.27 | 41.20 | 50.64 | 43.43 |
| | W4A4 | OS+ | 66.73 | 41.43 | 29.33 | 48.67 | 52.80 | 47.79 |
| | W4A4 | OmniQuant | 69.69 | 56.22 | 33.10 | 58.96 | 55.80 | 54.75 |
| | W4A4 | ZeroQAT | 71.86 | 58.27 | 32.68 | 57.16 | 56.35 | **55.26** |

Table F.1: Results of fine-tuning Llama1-7B on challenging MMLU benchmarks. 5-shot results are reported.

| Llama-7B (FP: 38.41%) | GPTQ | EfficientQAT | SmoothQuant | OmniQuant | ZeroQAT |
|---|---|---|---|---|---|
| W2A16 | 23.71% | 24.74% | - | 25.65% | 26.57% |
| W4A4 | - | - | 24.55% | 26.93% | 27.61% |

# F  EVALUATION ON MMLU

To demonstrate the generalizability of ZeroQAT in more realistic and challenging scenarios, we evaluate our method on MMLU, fine-tuning on the Alpaca dataset (Taori et al., 2023) and then evaluate. We conduct experiments based on Llama1-7B, the results are shown in Table F.1.

# G   THEORETICAL ANALYSIS

**Proposition 1** (Unbiasedness and explicit second-moment bound for the two-point ZO estimator)**.**
*Let $Q : \mathbb{R}^d \to \mathbb{Z}^d$ be the per-coordinate uniform quantizer of step size $\Delta > 0$ (rounding with
optional clipping/zero-point), and let $L(\cdot; B)$ be G-Lipschitz in its argument with respect to $\ell_2$:
$|L(z; B) - L(z'; B)| \le G\|z - z'\|_2$ for all $z, z'$ and all mini-batches $B$. For $\varepsilon > 0$ define the
Gaussian-smoothed (forward-only) objective*

$$f_\varepsilon(W) = \mathbb{E}_{u \sim \mathcal{N}(0, I_d)}\, \mathbb{E}_B\, L\big(Q(W + \varepsilon u); B\big),$$

*and the two-point ZO estimator with q i.i.d. directions $u_i \sim \mathcal{N}(0, I_d)$:*

$$g_b(W; B) = \frac{1}{q}\sum_{i=1}^{q} \frac{L\big(Q(W + \varepsilon u_i); B\big) - L\big(Q(W - \varepsilon u_i); B\big)}{2\varepsilon}\, u_i.$$

*Assume $\mathbb{E}_B\big[\,|L(Q(W + \varepsilon u); B)|\,\big] < \infty$ for all W and $\varepsilon > 0$. Then:*

(i) ***Unbiasedness.*** *The estimator targets the gradient of the smoothed objective:*

$$\mathbb{E}_{u,B}\big[g_b(W; B)\big] = \nabla f_\varepsilon(W).$$

(ii) ***Mean-squared error bound.*** *Writing the expectation over all randomness $(u, B)$,*

$$\mathbb{E}\,\big\| g_b(W; B) - \nabla f_\varepsilon(W)\,\big\|_2^2 \le \frac{1}{q}\bigg[2G^2\, d(d{+}2) + \frac{G^2\,\Delta^2\,d^2}{2\,\varepsilon^2}\bigg].$$

*In particular, ignoring the quantizer offset term (formally $\Delta{=}0$), the estimator's MSE
scales as $O\big(G^2 d^2/q\big)$ under standard Gaussian directions.*

*Proof. (i) Unbiasedness.* Let $U \sim \mathcal{N}(0, I_d)$ and write $Z = W + \varepsilon U$. Then $f_\varepsilon(W) = \mathbb{E}_{Z,B}\, L(Q(Z); B)$ with $Z \sim \mathcal{N}(W, \varepsilon^2 I_d)$. Differentiating under the integral with respect to the
mean of the Gaussian and using $\nabla_W \log p_{W,\varepsilon}(Z) = (Z - W)/\varepsilon^2$,

$$\nabla f_\varepsilon(W) = \mathbb{E}_{Z,B}\bigg[\frac{Z - W}{\varepsilon^2}\, L(Q(Z); B)\bigg] = \frac{1}{\varepsilon}\mathbb{E}_{U,B}\big[U\, L\big(Q(W + \varepsilon U); B\big)\big].$$

By antithetic symmetry of $U$,

$$\mathbb{E}_{U,B}\bigg[\frac{L(Q(W + \varepsilon U); B) - L(Q(W - \varepsilon U); B)}{2\varepsilon}\, U\bigg] = \frac{1}{\varepsilon}\mathbb{E}_{U,B}\big[U\, L\big(Q(W + \varepsilon U); B\big)\big],$$

hence $\mathbb{E}_{u,B}[g_b(W; B)] = \nabla f_\varepsilon(W)$.

*(ii) Second-moment/MSE bound.* Let

$$g(W; B, U) := \frac{L(Q(W + \varepsilon U); B) - L(Q(W - \varepsilon U); B)}{2\varepsilon}\, U.$$

Using independence of the $q$ i.i.d. samples, $\mathbb{E}\|g_b - \nabla f_\varepsilon\|_2^2 \le \frac{1}{q}\mathbb{E}\|g - \mathbb{E}g\|_2^2 \le \frac{1}{q}\mathbb{E}\|g\|_2^2$. By
$G$-Lipschitzness of $L(\cdot; B)$ and triangle inequality for $Q$,

$$\|g\|_2 \le \frac{G}{2\varepsilon}\big\|Q(W + \varepsilon U) - Q(W - \varepsilon U)\big\|_2 \|U\|_2 \le G\|U\|_2^2 + \frac{G\,\Delta\sqrt{d}}{2\varepsilon}\|U\|_2,$$

where we used the standard quantization geometry $\|Q(x) - Q(y)\|_2 \le \|x - y\|_2 + \|Q(x) - x\|_2 + \|Q(y) - y\|_2 \le \|x - y\|_2 + \Delta\sqrt{d}$ and $\|Q(z) - z\|_2 \le (\Delta/2)\sqrt{d}$. Applying $(a+b)^2 \le 2a^2 + 2b^2$
and Gaussian moment identities $\mathbb{E}\|U\|_2^2 = d$, $\mathbb{E}\|U\|_2^4 = d^2 + 2d$ yields

$$\mathbb{E}\,\|g\|_2^2 \le 2G^2\,\mathbb{E}\|U\|_2^4 + \frac{G^2\,\Delta^2 d}{2\varepsilon^2}\,\mathbb{E}\|U\|_2^2 = 2G^2\,\big(d^2 + 2d\big) + \frac{G^2\,\Delta^2\,d^2}{2\,\varepsilon^2}.$$

Dividing by $q$ completes the proof. $\qquad\square$

**What STE assumes and why it is biased.** The straight-through estimator (STE) replaces the ill-defined Jacobian $J_Q(W)$ of the piecewise-constant quantizer $Q$ by a hand-crafted surrogate $S(W)$ (e.g., $S(W) = I$ or a clipped indicator). The chain rule then yields the surrogate update

$$g_{\text{STE}}(W; B) = S(W)^\top \nabla_Q L(Q(W); B).$$

Because $Q$ is flat almost everywhere, the true chain rule gives $J_Q(W) = 0$ a.e., so asserting $J_Q(W) \approx S(W)$ implicitly enforces *gradient invariance to the discrete parameterization*: $\nabla_W L(Q(W); B) \approx \nabla_Q L(Q(W); B)$ regardless of whether small perturbations of $W$ actually change $Q(W)$. This mismatch makes $g_{\text{STE}}$ a biased estimator of any well-defined target (e.g., $\nabla f_\varepsilon(W)$ from Gaussian smoothing, or Clarke's generalized gradient of $f$), and the bias can remain large away from quantization thresholds where the true smoothed gradient vanishes in magnitude.

**Proposition 2** (Worst-case STE bias in expectation, 1-D). *Assume $d = 1$ and a uniform $b$-bit quantizer of step $\Delta > 0$. Let $L(z; B) = G z$ be a $G$-Lipschitz linear loss in its (quantized) argument. For $W \in \mathbb{R}$, let $r(W)$ be the distance to the nearest quantization threshold and set $t := r(W)/\varepsilon$. Consider the common STE choice $S(W) \equiv 1$. Then, for every $W$ and $\varepsilon > 0$,*

$$\left\| \mathbb{E}_B \big[ g_{\text{STE}}(W; B) \big] - \nabla f_\varepsilon(W) \right\| \geq G - \frac{G}{\sqrt{2\pi}} \Big( \tfrac{\Delta}{\varepsilon} + 2t + \tfrac{2}{t} \Big) e^{-t^2/2}.$$

*In particular, for any $\delta \in (0, 1)$, if*

$$t \geq \sqrt{2 \log \left( \frac{1}{\sqrt{2\pi}\,\delta} \Big( \tfrac{\Delta}{\varepsilon} + 2t + \tfrac{2}{t} \Big) \right)},$$

*then $\left\| \mathbb{E}_B[g_{\text{STE}}] - \nabla f_\varepsilon(W) \right\| \geq (1 - \delta)G$; i.e., the STE exhibits an $\Omega(G)$ bias in expectation away from thresholds.*

*Proof via two lemmas.* We first record two standard ingredients.

**Lemma 1** (1-D Gaussian tail identities). *If $U \sim \mathcal{N}(0, 1)$ and $t \geq 0$, then*

$$\mathbb{E}\big[|U| \mathbf{1}\{|U| \geq t\}\big] = 2\phi(t), \mathbb{E}\big[U^2 \mathbf{1}\{|U| \geq t\}\big] = 2\big(t\phi(t) + 1 - \Phi(t)\big), \mathbb{P}(|U| \geq t) = 2(1 - \Phi(t)), \tag{6}$$

*and Mills' bound $1 - \Phi(t) \leq \phi(t)/t$ holds for $t > 0$, where $\phi(t) = \frac{1}{\sqrt{2\pi}} e^{-t^2/2}$ and $\Phi$ is the standard normal cdf.* $\square$

**Lemma 2.** *Let $d = 1$ and $t = r(W)/\varepsilon$. If $L(\cdot; B)$ is $G$-Lipschitz in its argument, then*

$$\left\| \nabla f_\varepsilon(W) \right\| \leq \frac{G}{\sqrt{2\pi}} \Big( \tfrac{\Delta}{\varepsilon} + 2t + \tfrac{2}{t} \Big) e^{-t^2/2}.$$

Proof. *From the Gaussian-smoothing representation (two-point form),*

$$\nabla f_\varepsilon(W) = \mathbb{E}_{u,B} \left[ \frac{L(Q(W + \varepsilon u); B) - L(Q(W - \varepsilon u); B)}{2\varepsilon} u \right].$$

*By $G$-Lipschitzness and symmetry,*

$$\|\nabla f_\varepsilon(W)\| \leq \frac{G}{2\varepsilon} \mathbb{E}_u \big[ |u| \big| Q(W + \varepsilon u) - Q(W - \varepsilon u) \big| \big].$$

*If $|u| < t$, both perturbations stay in the same quantization cell and the difference vanishes; otherwise, the quantization geometry yields $\big| Q(W + \varepsilon u) - Q(W - \varepsilon u) \big| \leq (2\varepsilon|u| + \Delta) \mathbf{1}\{|u| \geq t\}$. Hence*

$$\|\nabla f_\varepsilon(W)\| \leq \frac{G}{2\varepsilon} \mathbb{E}[(2\varepsilon|u| + \Delta)|u| \mathbf{1}\{|u| \geq t\}].$$

*Expanding and applying Lemma 1 (with Mills' bound) gives the claim.* $\square$

We now prove the proposition. For the stated STE with $S(W) \equiv 1$ and the linear loss $L(z; B) = Gz$, one has $\nabla_Q L(Q(W); B) \equiv G$, hence

$$\mathbb{E}_B\big[g_{\mathrm{STE}}(W; B)\big] = G.$$

Therefore,

$$\left\| \mathbb{E}_B[g_{\mathrm{STE}}] - \nabla f_\varepsilon(W) \right\| \geq G - \left\| \nabla f_\varepsilon(W) \right\| \overset{\text{(Lemma 2)}}{\geq} G - \frac{G}{\sqrt{2\pi}}\left( \tfrac{\Delta}{\varepsilon} + 2t + \tfrac{2}{t} \right) e^{-t^2/2}.$$

Rearranging yields the thresholded $(1 - \delta)G$ lower bound.

**Remark 1.** *This formalizes the violation of gradient invariance and explains the $\Omega(G)$ expected bias away from thresholds. Multidimensional extensions follow by coordinate-wise threshold distances and union/tail bounds.*

