# OpenReview forum: "End-to-End On-Device Quantization-Aware Training for LLMs at Inference Cost"
_ICLR.cc/2026/Conference — ICLR 2026 Conference Withdrawn Submission_

### Official Review · Reviewer_Lzux · 2025-10-16

**Soundness:** 2
**Presentation:** 3
**Contribution:** 2
**Rating:** 4
**Confidence:** 5

**Summary:**

The paper proposes ZeroQAT, a zeroth-order (ZO), forward-only quantization-aware training framework for LLMs that supports both weight and activation quantization. It estimates gradients via two forward passes per random direction, removing backprop and optimizer state, and thus reducing memory to near-inference levels. The method jointly optimizes model weights with learnable activation smoothing (channel-wise scaling and shifting) and an adaptive weight quantizer (learnable step, zero-point, and asymmetric clipping). A lightweight variant freezes most parameters (pre-quantized) and keeps only attention Q/V in full precision to further cut memory for fine-tuning. Experiments on Llama-1/2 and OPT (up to 13B) show consistent improvements over representative  baselines, with on-device demos (OnePlus 12) for OPT-6.7B.

**Strengths:**

1. Clear idea and solid motivation: combining ZO with QAT to avoid STE bias/backprop memory while retaining end-to-end optimization.
2. Practical design: learnable smoothing and adaptive asymmetric clipping; lightweight Q/V-only variant for memory-limited fine-tuning; on-device evidence.
3. Writing is clear.

**Weaknesses:**

1. Scale and feasibility: As an efficiency-oriented QAT approach, the paper’s efficiency results suggest that running ZeroQAT on larger models (e.g., ≥30B, potentially even 70B) on a single A100 may be feasible. However, such larger-scale experiments are missing. Please add results or a careful quantitative feasibility analysis.
2. Model recency and difficulty: The benchmarks rely on older model families (Llama-1/2, OPT). It would strengthen the case to include modern, harder-to-quantize models (e.g., Llama-3 family, Qwen), which the community has found more challenging than Llama-1.
3. Fairness and clarity of comparisons: EfficientQAT natively supports Llama-2/3 under W2/3/4 A16 g64/128. In the Llama-2 tests (e.g., Table 3), EfficientQAT is not reported, while in the Llama-1 tests (e.g., Table 4) EfficientQAT appears to be evaluated under per-channel granularity rather than its recommended g64/128. This looks unfair. Please (a) add EfficientQAT to Llama-2 (and ideally Llama-3) under its native/recommended settings, and (b) avoid per-channel settings when the baseline’s recommended configuration is grouped.
4. Ablation on shifting: Compared to OmniQuant (which uses clipping and scaling), ZeroQAT additionally introduces a shifting parameter for activation smoothing. Given that ZeroQAT’s improvements over OmniQuant are sometimes modest in PTQ-style evaluations, please provide an ablation that disables the shifting parameter (keeping scaling and clipping) to isolate its contribution and verify whether ZeroQAT still provides a meaningful gain over OmniQuant-like corrections.

**Questions:**

1. Zeroth-order vs backprop accuracy: How much accuracy/loss gap does ZO-QAT have versus FO-QAT? While a full end-to-end comparison may be impractical, could you add a single-layer/block experiment comparing FO-QAT (with/without STE) and ZO-QAT under matched budgets to quantify the trade-off?
2. More tests and fair comparisons: Please expand model coverage (Llama-3 or Qwen), add larger models (≥30B) or provide a rigorous feasibility analysis, use EfficientQAT’s native g64/128 settings on Llama-2/3, and add the shifting ablation described above.

---

### Official Review · Reviewer_V9Pf · 2025-10-20

**Soundness:** 2
**Presentation:** 2
**Contribution:** 2
**Rating:** 2
**Confidence:** 5

**Summary:**

In this work, a new QAT scheme, called ZeroQAT, has been proposed for the weight-activation quantization.

To reduce the memory cost required for backpropagation during QAT, the authors employed the zeroth-order optimization for the gradient computation.

A lightweight variant of ZeroQAT has also been proposed for quantized fine-tuning.

**Strengths:**

The paper is well-written and easy to follow.

**Weaknesses:**

1. Limited contributions
 - As the authors pointed out in line 202, recent works have already combined the zeroth-order optimization with QAT under weight-only quantization. It seems that the authors extend such works to the weight-activation quantization scenario, which seems to be a marginal contribution. If the authors believe that their contribution is meaningful, then they need to clarify 1) the difficulty of extending existing methods to the weight-activation quantization and 2) how they overcome such difficulty in this work.
 - All the contents in Section 4.2 can be found in the existing literature. Eqs. (4) and (5) have already been presented in OmniQuant. Recent works, such as OSTQuant, even combined the scaling and rotation for outlier suppression and optimized the related parameters through end-to-end optimization. To conclude, the idea of scaling-based outlier suppression (Eq. (4)) and weight clipping (Eq. (5)), and the idea of exploiting end-to-end optimization are not novel and can be found in existing literature.

2. Lack of literature survey
 - The baseline schemes in this work are outdated. There have been lots of subsequent works since GPTQ, SmoothQuant, and OmniQuant.
 - For the weight quantization, several works aim to enhance GPTQ by incorporating cross-layer dependencies (e.g., BoA) and resolving cumulative error propagation (e.g., GPTAQ), which thus overcomes the problem that the authors pointed out.
 - For the outlier suppression, there have been many works that enhance SmoothQuant and OmniQuant (e.g., QuaRot, SpinQuant, OSTQuant, FlatQuant, etc). Among them, SpinQuant, OSTQuant, and FlatQuant optimize the transform-related parameters through the end-to-end optimization and thus overcome "Objective inconsistency" that the authors pointed out as the limitation of conventional works.
 - Due to the lack of a literature survey, the authors compared their method only with outdated schemes. Also, I think the performance of the proposed ZeroQAT is not so good in that the Wiki2 PPLs of BoA + QuaRot on Llama-7B and Llama-13B under 2-bit quantization are 9.812 and 8.341, respectively (see Table 12 in BoA paper), which are better than the results reported in Table 3.

**Questions:**

See Weaknesses.

Also, I want to point out that the authors need to use the terminology "layer-wise reconstruction loss" carefully. In fact, OmniQuant does not use layer-wise reconstruction loss in Eq. (1) but uses block-wise reconstruction loss (i.e., l2-norm of difference in the outputs of Transformer blocks).

---

### Official Review · Reviewer_JYzn · 2025-10-28

**Soundness:** 2
**Presentation:** 2
**Contribution:** 2
**Rating:** 2
**Confidence:** 5

**Summary:**

This paper proposes an efficient Quantization-Aware Training (QAT) framework that estimates gradients using forward-only finite-difference (FD) approximation, achieving stable and resource-efficient training even under low-bit quantization.

**Strengths:**

- Novel use of forward-only finite-difference for QAT, removing the need for backpropagation.
- Significantly reduces memory and computation cost, enabling on-device training.
- Integrates adaptive outlier smoothing for improved low-bit act. quant. stability.
- Simple, practical framework with clear implementation feasibility.

**Weaknesses:**

- The comparisons are mostly against outdated PTQ and QAT methods. For a fair and convincing evaluation, the paper should include more recent baselines such as ParetoQ, UPQ, and BitNet (for ternary/binary quantization). It would also be interesting to see whether the proposed method can achieve comparable or superior performance under the same settings as BitNet.

- The paper omits recent PTQ methods like BoA (https://arxiv.org/abs/2406.13474) and FlatQuant (https://arxiv.org/abs/2410.09426), both of which combine outlier smoothing and show strong low-bit performance. In particular, BoA combines outlier smoothing techniques such as Quarot and SpinQuant, achieving remarkably strong performance under low-bit quantization settings.

- The experiments rely on older models. Evaluating on LLaMA-3, Gemma, or Qwen would better demonstrate the method’s applicability to current-generation LLMs.

**Questions:**

Could the authors compare the proposed method with more recent QAT approaches and STE variants, providing detailed analyses of convergence speed, training stability, and memory efficiency?

**Details Of Ethics Concerns:**

.

---

### Official Review · Reviewer_zUMV · 2025-11-01

**Soundness:** 3
**Presentation:** 3
**Contribution:** 3
**Rating:** 2
**Confidence:** 3

**Summary:**

The paper introduces ZeroQAT, a zero'th order quantization-aware training ‌(QAT) method that achieves both inference-level efficiency, and QAT-level accuracy. The paper begins by a study on the weaknesses of post-training quantiozation (PTQ) and QAT, then presents ZeroQAT including learnable scaling/smoothing parameters. Additionally, a more memory-efficient version of the ZeroQAT is provided specifically for fine-tuning, where only Q and V tensors need to be trainable. Finally, comparison with PTQ and QAT methods show promising performance both in terms of accuracy and efficiency.

**Strengths:**

- The idea is interesting and very suitable for accurate memory-efficient compression.
- Motivation by multiple studies on PTQ and QAT methods.
- Including the extra efficient variation for fine-tuning.

**Weaknesses:**

1. The runtime comparison is not convincing to me (please see questions for more details).
2. The experiments are missing some relevant baselines (again, see questions for more details).
3. More ablation on the zero'th order gradients are needed.

**Questions:**

1. While per-update runtime is reported in Table 7, I believe this is not enough. From my understanding, the zero'th order gradients should be noisy estimators of the true gradients. This added noise should potentially delay the convergence, which means more updates might be needed for ZeroQAT to converge. I think a study on this would be important, e.g., a loss curve during training and comparing it with other QAT methods.

2. Generally, while zero'th order gradients are the main component of the method, not much ablation has been done on their accuracy. For example, to what extent do they recover true gradients (let's say in terms of cosine similarity), or what happens end-to-end if ZeroQAT uses true gradients instead of zero'th order gradients?

3. Regarding the PTQ experiments, I believe some important and strong recent baselines are missing from the comparisons in Table 3. For example QuaRot [1], SpinQuant [2], and FlatQuant [3].

4. In Equation 4, I don't understand how the term $B+ \delta W$ is handled in practice. First, note that $B$ is a vector while $W$ is a matrix. Do you need to broadcast $B$? And second, I assume $B+ \delta W$ is a matrix that is not quantized, with the same shape as $W$. Does that mean you need to store it for inference, along with the quantized $\bar{W}$? If that's the case, the memory benefits will be partially lost during inference (as in you don't only need to store the quantized weights), so I assume I'm misunderstanding something. Please correct me if I'm wrong.

5. Maybe relevant for a discussion in the paper: LoTA-QAF [4] does ternary quantized fine-tuning.

I generally think the idea is interesting and could have impact on the field, but I don't think the current set of experiments/ablation are comprehensive enough. I would be open to increasing my score depending on how the rebuttal goes.

[1] https://arxiv.org/pdf/2404.00456
[2] https://arxiv.org/pdf/2405.16406
[3] https://arxiv.org/pdf/2410.09426
[4] https://arxiv.org/pdf/2505.18724

---

### Note · Authors · 2025-11-12

I have read and agree with the venue's withdrawal policy on behalf of myself and my co-authors.